# How Central Is the Domestic Pig in the Epidemiological Cycle of Japanese Encephalitis Virus? A Review of Scientific Evidence and Implications for Disease Control

**DOI:** 10.3390/v11100949

**Published:** 2019-10-15

**Authors:** Héléna Ladreyt, Benoit Durand, Philippe Dussart, Véronique Chevalier

**Affiliations:** 1Epidemiology Unit, Laboratory for Animal Health, French Agency for Food, Environmental and Occupational Health and Safety (ANSES), University Paris-Est, 94700 Maisons-Alfort, France; helena.ladreyt@anses.fr (H.L.); benoit.durand@anses.fr (B.D.); 2Agricultural Research for Development (CIRAD), UMR ASTRE, F-34090 Montpellier, France; 3Virology Unit, Institut Pasteur du Cambodge, Institut Pasteur International Network, PO Box 983, Phnom Penh 12201, Cambodia; 4Epidemiology and Public Health Unit, Institut Pasteur du Cambodge, Institut Pasteur International Network, PO Box 983, Phnom Penh 12201, Cambodia; 5Agricultural Research for Development (CIRAD), UMR ASTRE, Phnom Penh 12201, Cambodia

**Keywords:** Japanese encephalitis virus, pig, epidemiology, control

## Abstract

Despite the existence of human vaccines, Japanese encephalitis (JE) remains the leading cause of human encephalitis in Asia. Pigs are described as the main amplifying host, but their role in JE epidemiology needs to be reassessed in order to identify and implement efficient control strategies, for both human and animal health. We aimed to provide a systematic review of publications linked to JE in swine, in terms of both individual and population characteristics of JE virus (JEV) infection and circulation, as well as observed epidemiological patterns. We used the Preferred Reporting Items for Systematic Reviews and Meta-Analyses (PRISMA) statement to select and analyze relevant articles from the Scopus database, 127 of which were included in the review. Pigs are central, but the implication of secondary hosts cannot be ruled out and should be further investigated. Although human vaccination cannot eradicate the virus, it is clearly the most important means of preventing human disease. However, a better understanding of the actual involvement of domestic pigs as well as other potential JEV hosts in different JEV epidemiological cycles and patterns could help to identify additional/complementary control measures, either by targeting pigs or not, and in some specific epidemiological contexts, contribute to reduce virus circulation and protect humans from JEV infection.

## 1. Introduction

Japanese encephalitis (JE) is a serious vector-borne zoonosis and probably the most important cause of human viral encephalitis in South East Asia (SEA). In 2011, the last attempt to estimate the overall incidence of JE reported an approximated incidence of 67,900 JE cases in 24 Asian and Western Pacific countries. Approximately three-quarters of these concerned children, and JE remains a substantial public health issue even in areas that have developed human vaccination programs [1]. The fatality rate can reach 30%, and 30% to 50% of survivors may continue to suffer definitive neurological or psychiatric sequelae [2]. There are no licensed anti-JE drugs available, and the management of patients remains symptomatic. 

JE is caused by a flavivirus, which is part of the JE virus (JEV) serocomplex along with West Nile Virus (WNV), Murray Valley encephalitis virus (MVEV), Saint Louis encephalitis virus (SLEV), and Usutu virus (USUV). JEV distribution has long been limited to SEA but has now extended to Australia, Papua New Guinea, and a human indigenous case was recently confirmed in Africa [3]. To date, five genotypes (GI to GV) have been described, with most isolated strains belonging to genotypes I, II, and III [4]. While genotype III (GIII) was the most frequently isolated genotype throughout most of Asia from 1935 until the 1990s, a genotype shift occurred in the last 30 years. In pigs, genotype I (GI) started to be the most isolated genotype after the 90s in Japan, South Korea, India, Nepal, Thailand, Vietnam, and Cambodia [5,6]. 

JEV is transmitted from animals, especially pigs, to humans by *Culex* mosquitoes, such as *Culex tritaenyorhynchus* or *Culex gelidus*, and probably by some *Aedes* mosquitoes [7,8,9,10,11,12,13,14,15]. The commonly described cycle implicates Ardeid birds as JEV reservoirs, pigs as the main amplifying hosts, and *Culex* mosquitoes as vectors [16,17,18,19,20]. In addition to vector-borne transmission, recent findings suggest that direct transmission between pigs could also occur [21,22,23]. Domestic birds might also be involved in the cycle. Indeed, they were shown to be exposed to JEV and could develop sufficient viraemia to re-infect mosquitoes when bitten [24,25,26,27]. Other animals, such as cattle or dogs, were shown to be exposed to JEV, but no study investigated their potential role in the epidemiological cycle, and they are, for now, considered to be dead-end hosts [28,29,30,31,32,33,34,35]. Humans and horses, also subjected to developing fatal encephalitis, are, for now, only known to be dead-end hosts [36,37]. 

Since there is no antiviral treatment or antiviral prophylaxis, human vaccination remains the only available tool to protect humans from JEV infection [38], but it does not prevent JEV circulation. As a matter of fact, human JE cases still occur in countries where mass vaccination campaigns are implemented. If pigs are the main amplifying host of JE, breaking the mosquito–pig transmission cycle should stop virus circulation and protect humans from JEV infection. Thus, improving our knowledge on the characteristics of swine infection by JEV in terms of viraemia, the immune response, mechanisms of transmission, and clinical signs is necessary to better assess the role of domestic pigs in the epidemiological cycle, and to identify additional control measures focusing on pigs. How prevalent is JEV in swine? What are the different JE epidemiological patterns? Do pig-targeted control measures allow control of JEV circulation? The present review aimed at synthesizing the knowledge related to pigs and JEV, at both the individual and population levels, in order to discuss the importance of swine in the JEV transmission cycle. The role of pigs in the JEV epidemiological cycle is described based on the available studies, and potential pig-related control measures are discussed.

## 2. Materials and Methods 

### 2.1. Protocol, Search Process, and Databases

This review followed the Preferred Reporting Items for Systematic Reviews and Meta-Analyses (PRISMA) method for systematic reviews and meta-analyses [39]. All studies dealing with both JEV and swine were eligible for the systematic review. Serological surveys, experimental and ecological studies, and JEV outbreak investigations were considered. We only reviewed articles written in English and no publication date restriction was imposed, thus, the publication year extended from 1947 to 2019. The Scopus database was searched electronically using the following request: (“Japanese encephalitis” or “Japanese B encephalitis”) and (“pigs” or “swine”). We used the “all field” option in order to collect the articles in which the search terms appeared in the titles, abstracts, or keywords. 

The same person conducted all of the initial searching and screening. Articles were listed in an Excel file to sort them and keep track of any excluded ones. Duplicates were removed and the study selection was made in three steps: i) title screening, ii) abstract reading of papers kept after title screening, and iii) full-text reading of papers kept after abstract reading. 

### 2.2. Inclusion and Exclusion Criteria 

The study selection was based on the following inclusion criteria: English-written articles reporting cross-sectional or longitudinal serological surveys, experimental infections, description of clinical disease, pathogenicity, transmission routes, viraemia, immune response and diagnosis, effects of vaccination, control measures, epidemiological cycle and systems, and all subjects being directly related to swine. The exclusion criteria included reviews, notes and reports from congresses, travelers’ recommendations, genome sequencing alone, experimental virology alone (viral structure, virus–cell interactions), development of diagnostic tests, not focused on swine (i.e., focused on humans, mosquitoes), human outbreak notifications, focus on viruses other than JEV, and mathematical modeling approaches. 

## 3. Results

### 3.1. Study Selection

The query of the Scopus database was performed on 15 March 2019 and returned 667 records. The study selection process is represented in Figure 1. At the end of the selection process, 127 studies were included in the qualitative synthesis.

### 3.2. Individual Characteristics of JEV Infection in Swine

All mentioned virologic and serological tests are described in Appendix A [40,41,42,43,44,45,46].

#### 3.2.1. Viraemia in JEV-Infected Pigs

Viraemia is the presence of virus in the blood. During this phase, JEV can be transmitted to mosquitoes that bite the viraemic pig if viraemia level is high enough. It was experimentally shown that viraemia levels of about 10^4^ infectious units per mL appeared to be sufficient to transmit the virus to mosquitoes [47,48,49,50]. Six articles reported experimental works related to viraemia in JEV-infected pigs. Sows and piglets were inoculated with different JEV strains and bled daily in order to detect the virus, by intracerebral inoculation of suckling mice (ICISM) in the earliest studies, and JEV ribonucleic acid (RNA) by reverse transcription-polymerase chain reaction (RT-PCR) in the more recent ones. The results are presented in Table 1. Depending on the study, viraemia was detected between one and five days post-infection (dpi), and lasted from three to five days [22,47,51,52,53,54]. One study concluded that the higher the concentration of inoculated virus, the earlier the virus was detected (as early as one day). When detected, virus titers reached 2.6 log LD_50_/0.03 mL of blood upon titration intracranially in weanling mice, with LD_50_ being 50% of the lethal dose [47]. More recently, viraemia was quantified by detecting JEV RNA using quantitative RT-PCR (RT-qPCR) on the inoculated piglets, reaching about 10^4^ RNA units/mL [22,54]. 

Neither of these measures of viraemia can be quantitatively compared, as infectivity assays detect the presence of infectious virions whereas RT-qPCR detects RNA from both non-defective and defective virions [53]. 

These experimental studies consistently showed that JEV viraemia was early and short. This may explain why JEV is difficult to isolate from pigs under field conditions in cross-sectional studies as well as in longitudinal studies if the periodicity of blood sampling is not high enough. In comparison with other pig diseases of importance, African Swine Fever (ASF) causes viraemia for three weeks to up to two months [55,56,57] and Classical Swine Fever (CSF) for one to three weeks [58,59]. 

Viraemia characteristics under field conditions could be documented through the follow-up of pigs and the detection of JEV in blood. Ueba et al. followed up piglets for several months, and tested them from once a day to once every four days using ICISM. The results confirmed the viraemia durations observed under experimental conditions, as viraemia was detected for three to five days [60]. 

#### 3.2.2. Humoral Response of Pigs after JEV Infection

The humoral response after experimental infection was studied in six articles (Table 1). Depending on the protocol of the study (JEV strain, inoculation route and dose, age of the piglets), antibodies after JEV experimental infection appeared at 3 to 7 dpi, and were detected until the end of the study (7–35 dpi). Ueba et al. reported that viraemia dropped as soon as hemagglutination inhibition assay (HIA) antibodies were detected [60]. This suggested that the viraemia was relatively short due to the early onset of the immune response. In the 90s, Geevarghese et al. monitored piglets under real conditions for a longer time and showed a persistence of antibodies detected by HIA for up to three years [61], although this long-lasting immunity may have been due to repeated exposure to JEV, thereby boosting immunity.

#### 3.2.3. Clinical Signs

Clinical signs, either in sows or in piglets after experimental virus inoculation or after natural infection, were described in fourteen articles. Detailed methodology of the surveys and their results are provided in Table 3.

One experimental work described JEV-associated sow reproductive failure (Table 2) [51]. Depending on the JEV strain used for sow inoculation, the consequences of experimental infection of seronegative sows ranged from no impact on the litter to reproductive disorders in two out of six sows. Reproductive failure was characterized as presence of mummified or hydrocephalic fetuses in the litters. In both cases, sows developed viraemia and an immune response after infection. 

It was shown that experimentally JEV-infected piglets also developed clinical signs. Results of the four concerned studies are summarized in Table 2. Fever was detected one day post-infection and mild neurological signs, such as hind limbs tremor or depression, were observed until a maximum of 10 dpi. All clinical signs disappeared in few days without treatment. No macroscopic lesions were detected, but three articles reported unspecific microscopic lesions on inoculated piglets. 

The clinical impacts of JEV are more difficult observe under field conditions. In seven articles, authors tested tissues of aborted of stillborn piglets in herds with apparent reproductive failure and in which JEV was suspected. These results are summarized in Table 3. JEV was isolated in 5/37 to 8/8 of the brains of aborted or stillborn piglets. Four articles also gave quantitative information about sow reproductive failure (i.e., the presence of stillborn piglets or aborted or mummified fetuses in one litter) in the affected herds. The authors identified reproductive failure rates ranging from 15% to 36% of the reproductive sows [50,64,65,66]. Another study based its analysis on breeders’ interviews in a herd where anti-JEV antibodies were detected in sows. Information related to reproductive performances were collected (number of piglets born in total and alive in the last litter, occurrence of abortion and birth of stillborn or weak born piglets with or without neurological symptoms). According to farmers, in the first herd, 31 out of 51 sows showed reproductive disorders. The authors then showed a positive correlation between the detection of anti-JEV antibodies (based on Enzyme-Linked Immunosorbent Assays detecting IgG (IgG ELISA) results) and the number of stillborn piglets for sows younger than 1.5 years old [67], suggesting that JEV infection caused both reproductive failure and the appearance of protective antibodies in young pregnant sows. 

Two field studies reported clinical signs in piglets potentially induced by JEV. In 2009, piglets developed viral encephalitis and died in a farm in Japan. Seven brains were sampled, in which JEV was detected by RT-PCR [71]. In 2014 in India, a study was conducted in pigs with a history of reproductive failure. Macroscopic and microscopic lesions were detected in association with JEV infection in stillborn piglets, confirmed by RT-PCR [65]. Stillborn piglets showed subcutaneous hemorrhages, hydranencephaly, or swollen brains with dilatation of the ventricular spaces and thinning of the surrounding parenchyma. Histopathology of brain tissue revealed widespread edema, congestion, microhaemorrhages in parenchyma, neuronal degeneration, and accumulation of glial cells. These microscopic lesions were in accordance with the experimental work presented above. 

Finally, JEV might cause reproductive problems in boars, but studies are limited. Ogasa et al. detected JEV in the semen of two out of five experimentally inoculated boars by ICISM, and reported a reduced spermatozoal motility and concentration in two other boars [72]. On the other hand, in Teng et al. two out of twelve diseased boars showing testicular swelling tested positive by RT-PCR on seminal fluid [69].

In conclusion, both experimental and field studies showed that the JEV clinical signs mainly consisted of reproductive disorders in females (and to a lesser extent in males), ranging from weak piglet births to abortions. JEV was detected in the brain tissues of infected piglets that developed unspecific signs of encephalitis.

#### 3.2.4. Anti-JEV Maternal Antibodies 

The persistence of anti-JEV maternal antibodies in piglets under field conditions was described in four articles. The results are presented in Table 4. All four studies were based on the follow-up of piglets in Japan, Cambodia, and South India. In early studies, anti-JEV maternal antibodies disappeared in piglets aged 1.5 to 4 months [51,73]. These results were confirmed by recent studies that showed a disappearance of anti-JEV maternal antibodies in piglets aged between 2 and 3.5 months [74,75]. 

#### 3.2.5. JEV Excretion

Few studies analyzed JEV excretion. Three studies focused on JEV oro-nasal shedding, which may explain the diffusion of JEV in pig herds where they are no or few mosquitoes. Piglets were experimentally infected and monitored for clinical signs, viraemia, excretion, and virus tropism. Viral shedding in nasal secretions was detected by RT-qPCR from 2 to 8 dpi at levels similar to viraemia, i.e., from 10 to 10^4^ units/mL, depending on the study [22,23,54]. Authors also reported a prolonged detection of JEV RNA in the tonsils, until at least 28 dpi. 

No studies looked for JEV excretion in abortion fluids, whereas the virus was detected in brain tissues of aborted or stillborn piglets (see Table 5). Ricklin et al. detected JEV RNA in urine at a low frequency (in one urine sample of the 28 monitored piglets) [21], and authors suggested tropism differences between JEV and other flaviviruses, such as WNV or dengue virus (DENV) [76,77].

One study detected JEV in seminal fluids of diseased wild boars [69].

### 3.3. JEV Transmission Mechanisms and Patterns and Pig-Related Control

#### 3.3.1. JEV Transmission Mechanisms 

To date, two transmission mechanisms of JEV to and between pigs have been studied, i.e., the well-known vector transmission and potential direct transmission through oro-nasal shedding. 

##### JEV Vector-Borne Transmission

In 1964, an experimental study analyzed the mosquito-pig and the pig-mosquito transmission [78]. Authors inoculated *Cx. tritaeniorhynchus* mosquitoes with JEV and let them bite pigs with varying anti-JEV antibody titers. Pigs were bled daily before and after their contact with the infected mosquitoes. ICISM tests were performed on blood samples (intracranial inoculation to 3-day-old mice) and virus titer was expressed as a LD_50_. JEV was detected in all pigs except from the one that had the highest anti-JEV antibody titer, showing that the mosquito–pig transmission was effective. While the virus was detected in pigs (during the viraemic period), healthy mosquitoes were allowed to feed on them. Fourteen days later, authors showed the presence of JEV in salivary secretion of these mosquitoes. Anti-JEV antibody-free pigs presented the highest viraemia (log 1.1–2.5 LD_50_) and re-infected 10%–96% of the mosquitoes feeding upon them. These results were confirmed a few years later [52].

More recently, in China, the genomes of JEV strains isolated from pigs and mosquitoes were found to be very similar, demonstrating JEV vector-borne circulation under field conditions [70]. 

##### JEV Direct Transmission

Besides vector-borne transmission, the three experimental studies showing nasal and oral shedding of JEV presented above (3.2.5. JEV excretion) suggested a risk of direct transmission between pigs [22,23,54]. Table 5 presents the results of two additional studies that showed the effectiveness of this direct transmission. Authors first inoculated piglets with known doses of JEV and put them in contact with naïve ones in absence of mosquitoes [21]. Then, piglets were orally and intra-nasally inoculated with known doses of JEV and monitored [21,63]. In both studies, clinical signs were observed and viraemia and oro-nasal shedding of JEV were detected by RT-PCR. The incubation period for pigs infected by contact was three to five days with respect to the development of viraemia. The incubation period for pigs oro-nasally infected ranged from one to three days depending on the inoculated doses. Viraemia reached 10^3^ to 10^4^ U/mL, with 1 U corresponding to the RNA quantity found in one 50% tissue culture infective dose (TCID_50_) of a virus preparation they used.

Furthermore, a recent study suggested that nasal epithelium could be a route of entry and exit for JEV in pigs. As JEV is shed to both the apical and basolateral sides of the epithelial cells, such an infection could mediate virus entry into the host as well as oro-nasal virus spread to other hosts in a manner principally comparable to that of respiratory viruses such as influenza virus [79].

#### 3.3.2. Geographical Distribution of JEV in Swine

From 1966 to 2016, the presence of anti-JEV antibodies was reported in swine in seventeen Asian and/or Pacific countries, from India in the west to Japan in the east and from the north of Australia in the south to China in the north (Figure 2). However, since the antibodies detected by either ELISA or HIA might have corresponded to those of other viruses in the same serological complex with JEV, only seroneutralization tests, such as the plaque reduction neutralization test (PRNT), allow the confirmation of true JEV exposure. In some studies, a good consistency between high HIA titer and 50% plaque reduction titer was observed [74,75,80].

Table 6 summarizes the most recent cross-sectional and longitudinal serological surveys undergone in swine for each country, with older references being indicated in the last column. 

Globally, the prevalence levels ranged from 3.1% to 74% (in Table 6, the prevalence was only calculated if more than 100 swine were tested). In the most recent studies (pigs sampled after 2007), the highest seroprevalences in domestic pigs (>50%) were found in Laos, Cambodia, Vietnam, and India. In Cambodia (2007, [80]) and in Thailand (1983, [81]), seroprevalence detected by HIA was shown to increase with age once piglets had loss their maternal antibodies. However, in Sri Lanka in 1988, authors did not observe any significant correlation between the ages of the sampled animals and the seroprevalence detected by the reference seroneutralization test (SNT) [82]. 

Few seroprevalence data were reported for China, even though the virus was detected in pigs by RT-PCR on several occasions [64,68,69,70,71]. However, in Tibet, three studies reported IgM prevalence levels using capture ELISA ranging from 5% to 33% of the sampled pigs. JEV infection was confirmed by RT-PCR in some of the positive animals, suggesting a relatively high level of the incidence of JEV infection in Tibetan pigs. 

Concerning wild boars and feral pigs, high JEV seroprevalence levels were found in Japan (between 44% (*n* = 117) and 83% (*n* = 36)), South Korea (66% (*n* = 288)), and Singapore (100% (*n* = 68), last study in 1999). In 1998, anti-flavivirus antibodies were detected by HIA in almost 80% of feral pigs in the north of Queensland, Australia [83]. These feral pigs were sampled during a JE outbreak investigation, suggesting that some of these anti-flavivirus detected antibodies were anti-JEV antibodies and that there may be a link between wildlife and JE circulation. Only SNT tests could confirm this statement. In other countries, the comparison between JEV seroprevalence levels observed in domestic pigs and in wild boars is difficult; in Japan, wild boars and domestic pigs were sampled from different islands, and in South Korea, only wild boars were sampled. In Singapore, however, wild boars were sampled in the same year and the same area as domestic pigs, and both seroprevalences were high (respectively 100% (*n* = 28) and 94% (*n* = 81)). 

#### 3.3.3. Epidemiological Patterns 

Results of several field studies suggested that either the peak of mosquito abundance or the rise of infected mosquitoes (detection of JEV by ICISM) coincided with the seroconversion period of the pigs (detected by an increase of the proportion of positive pigs for anti-JEV antibodies using HIA) [73,102,117,118,119] and was linked to the occurrence of clinical signs in humans [14,78,82,113,120,121]. These latter results raised the question of the seasonality of JEV transmission (see Figure 6 in Konno et al. reviewed in Van den Hurk et al. [120,122]).

Seasonality of JEV circulation in pigs was shown either by longitudinal serosurveys (in which pigs were followed-up), or by repeated cross-sectional serosurveys in slaughterhouses or on farms. In the longitudinal studies, depending on the protocol, 2- to 4-month-old piglets were put into pens and blood sampled daily to monthly for four months to two years. In the longest surveys, the piglets were replaced when they seroconverted [14,73,74,75,78,81,83,117,123,124,125]. In repeated cross-sectional surveys, pigs aged from four to twelve months were sampled every month in slaughterhouses over one to ten years, depending on the protocol [50,101,102,112,114,118,120,121,126,127,128], and reproductive sows were sampled monthly on a farm in south Vietnam [67]. These results are presented in Table 7. 

Three main epidemiological patterns can be distinguished:(i)An endemic pattern with no seasonality, where JE circulation is high all year round, as in Cambodia where two longitudinal surveys showed that all monitored piglets seroconverted in less than four months, regardless of the time of year [74,75].(ii)An endemic pattern with seasonality, where JE circulation also occurs all year round but peaks during the hot and rainy seasons, as in north Vietnam [112,114], Laos [101,126], Malaysia [102], Indonesia [125], Thailand (except for the mountainous Chang Mai district) [14,123], and Taiwan [73,78,116].(iii)An epidemic pattern, with peaks of JEV transmission separated by periods of non-detection, as in north Australia [83], north India [117], Sri Lanka [82], the Chang Mai district in Thailand [124], and Japan [50,118,120,121,127,128]. In Japan, Konno et al. detailed cyclic outbreaks of JE among swine and human populations linked to vector abundances, which was reviewed in 2009 [120,122].

For the two latter epidemiological patterns, periods of high JE circulation were identified thanks to longitudinal studies; these are given in Table 7. Usually, JEV seroconversion rates peaked during the hot and rainy season. Longitudinal studies showed that during high circulation periods, almost all of the monitored piglets seroconverted against JEV from as early as one week (in Taiwan) to five and a half months (in Indonesia). 

In the 90s, in north Vietnam, two studies showed periodic variations of JEV seroprevalence in pigs (higher circulation in the wet and hot season), suggesting a seasonal circulation of JEV [112,114]. These results contrasted with those obtained in south Vietnam (Mekong delta), where a recent study did not identify any correlation between JEV seroprevalence in sows and season [67]. In Thailand, JE seasonality was reported to be much more marked in the Chiang Mai district [124], 700 km north from Bangkok, than in Bangkok; JE circulation in pigs was detected only from March to December (reaching 90% of JEV seroconversion between May and July), while circulation was detected all year round in Bangkok district [78,123]. In Taiwan, it is believed that the circulation of JE also occurs in winter, since Chan et al. detected HIA antibodies in 15.6% of 5-month-old pigs raised in winter, suggesting that this prevalence was not of maternal origin but due to a slight circulation [116].

Besides these temporal patterns, the intensity of JEV circulation also varies with the landscape. *Cx. tritaeniorhynchus*, *Cx. gelidus*, and *Culex quinquefasciatus*, the main vectors of JEV, are rural mosquito species that are mainly distributed in rice field agroecosystems of Asian countries, flooded either by rain or irrigation [11,122,130,131,132,133,134,135,136,137]. For this reason, JEV seroprevalence studies were mainly conducted in swine reared in (or close to) such ecosystems. In such contexts, JEV seroprevalence was shown to be potentially very high, with values ranging from about 40% in Indonesia to more than 75% of HIA- or ELISA-positive results (confirmed by SNT) in Hong Kong or Laos [31,34,84,85,89,101,103,104,116]. These high prevalence levels led to JE being considered mainly a rural disease, and proximity to rice fields and pig rearing, particularly backyard farming, were identified as major risk factors of JE in humans [89]. 

However, several studies conducted in Taiwan [116], Thailand [14,81], Hong Kong [84], Japan [128], and more recently in Cambodia [74,75], Vietnam [113,115], and Malaysia [31] showed that JEV can also circulate in swine in peri-urban or even urban areas. The JEV seroprevalence values ranged from 26.5% to more than 90% of HIA- or IgG ELISA-positive results in cross-sectional studies, and seroconversion rates reached 100% of the monitored piglets in longitudinal studies performed in Cambodia. These observations suggest the implication of other vectors and/or other host species in the corresponding epidemiological systems.

#### 3.3.4. Pig-Related JEV Control

Pig-related control measures have already been reviewed elsewhere [20,138,139]. Both JEV inactivated and live-attenuated vaccines derived from cell cultures are used, and pig vaccination has been reported in Japan, Nepal, Taiwan, and South Korea, but to a limited extent and only in order to protect pregnant sows from reproductive disorders [140]. Human vaccines are widely used [141,142], and mass vaccination program in humans were put into place in several countries, such as Japan, South Korea, and Taiwan, where programs are long-lasting and of high quality, or Nepal, Malaysia, India, Sri Lanka, Thailand, Vietnam, and China, where programs are emerging [1]. In the animal sector, vaccine development and vaccine program implementation are more limited [48,143,144], except in South Korea where a vaccination program in swine with a live-attenuated vaccine has been implemented over the last 30 years, reducing the prevalence of the disease in pigs [145,146].

In our selection, two articles reported the impact of JEV vaccination on the frequency of reproductive failure in sows. The effectiveness of JEV vaccination on reproduction was first shown in 1971 in the field by monitoring vaccinated and unvaccinated groups of sows. The vaccinated group presented two times less abnormal farrows (partial or total abortion or stillbirths), fewer malformed or stillborn piglets (1%–7% against 31%–54% in the unvaccinated group), and larger litters (two to three more piglets by litter) [147]. In experimental conditions, vaccination protected against fetus mummification [49].

Three surveys suggested that JEV pig vaccination interfered with the JEV epidemiological cycle, thereby reducing the impact of JE on public health. A live-attenuated vaccine with Freund’s complete adjuvant showed a drastic decrease in viraemia in piglets [60,148]. A few years later, the live-attenuated vaccine was confirmed to reduce viraemia and prevent mosquito re-infection in pigs [48,147]. However, the impact of mass vaccination in pigs on human disease has not been demonstrated yet. In South Korea, after 30 years of the JEV vaccination program in swine, outbreaks in humans were still not prevented [145].

## 4. Discussion

This review confirmed that pigs are central in JE epidemiology, not only for virus maintenance and amplification, but also in transmission to humans. Pigs develop high levels of viraemia that last for two to four days and attract JE vectors, and the rapid turnover of piglets in any kind of pig farming induces a permanently high proportion of susceptible individuals that facilitate JEV circulation. Moreover, pig farms are often located close to human dwellings, especially with backyard farming, which is common in Asia; this proximity facilitates human infection. 

From 1966 to 2016, the presence of anti-JEV antibodies was reported in swine in seventeen Asian and/or Pacific countries, with seroprevalence levels ranging from 3.1% to 74%. Although the diversity of study areas, periods, and protocols partly explain this wide interval, the reported seroprevalence levels are difficult to compare due to the variety of the serological tests used and the lack of specificity of some of these tests. Indeed, the reference technique for the serological diagnosis of flavivirus infections is SNT [122]. Only a few cross-sectional surveys selected in this review used SNT to confirm that the flavivirus they detected (either by HIA or ELISA) was JEV [34,35,82,96,102,105,111]. In the remaining studies, authors used HIA or IgG ELISA. HIA exploits the ability of viral envelope proteins to aggregate erythrocytes in the absence of neutralizing envelope antibodies, and is subject to many cross-reactivities in the JEV serocomplex and with other flaviviruses due to the type of antigen used [149]. Similarly, ELISA tests, which are based on a colorimetric reaction for which the color intensity is related to the antibody concentration, are specific to flaviviruses but not exclusively to JEV, depending of the type of antigen used [150]. Moreover, although all of the selected HIA studies used the same technique [149], many ELISA kits are available and were used in the published surveys. Flaviviruses other than JEV, such as WNV and DENV, are known to circulate in JEV-infected countries [151]. This co-circulation poses a diagnostic challenge due to antibody cross-reactivity within and between the different serocomplexes [53,152,153,154,155,156,157,158,159]. Thus, confirming HIA- and ELISA-positive results via SNT appears essential in the proper evaluation of JEV seroprevalence [160]. It is also worth noting that, although it is the reference test, SNT may be subject to limited cross-reactions within the serocomplex of Japanese encephalitis. This sometimes requires testing other flaviviruses (such as WNV and DENV) in parallel to JEV, and the implementation of a decision algorithm based on the SNT titers, in order to identify the flavivirus to which the animals were exposed [150]. This complex confirmation procedure was used in few selected studies [34,82,111]. High-quality data on the seroprevalence and incidence of JE are thus lacking in various countries, and there is a real need for research efforts into virological and serological methods for diagnosis and monitoring of JE.

JE is often described as a significant cause of reproductive disorders in sows and boars. Only one experimental study provided evidence of the impact of JEV infection on sows [51]. The reproductive impact of JEV appears difficult to quantify under field conditions, and statistical correlations between JEV infection and reproductive failure are rarely investigated. Moreover, the frequency of reproductive disorders appears to be linked to the epidemiological pattern of JEV. In endemic areas with no marked seasonality, year-round contact of swine with the virus leads to immunity in most gilts before sexual maturity; JEV infection in pigs is consequently of minor importance on reproductive performance [67]. Under seasonal transmission conditions, sows are more likely to present reproductive disorders if they are still susceptible when they reach sexual maturity. These observations might affect the pig-related control strategies that are discussed below. 

Several experimental studies showed that domestic pigs developed high viraemia, allowing for pig–mosquito–pig transmission [22,47,54]. Several surveys showed that the primary JEV vectors *Cx. tritaeniorhynchus*, *Culex vishnui*, and *Cx. gelidus* have a trophic preference for cows and pigs [161,162,163,164,165]. However, the two former species often showed higher rates of blood feeding on pigs than on cows, probably due to their plasticity to host availability, since in rural areas, pigs are often more abundant than cows [166]. Moreover, the number of susceptible pigs is always large due to a large turnover stemming from the breeding system. Finally, pig farming in Asia is often backyard breeding, which situates pigs very close to human dwellings. All these elements support the major importance of pigs in the JEV transmission cycle. 

However, recent studies suggested that other JE epidemiological systems may exist with small or no implications resulting from the domestic pig. First, autochthonous human JEV cases occurred in Seoul, South Korea, even though no pigs are reared in the city [26]. In 2013, in China, Teng et al. isolated JEV in mosquitoes, humans, and pigs, and showed that the same strain was identified in mosquitoes and human, but not in pigs [69]. Finally, two studies, one being very recent, showed that JEV continued to circulate decades after the abolition of pig farming on Singapore Island [35,167]. 

It is generally considered that Ardeid birds, such as egrets and herons, are the wild reservoir hosts [168,169,170,171,172,173]. However, little recent evidence of this statement is available [18,168,174]. Several surveys or observations showed that other species, such as domestic birds, are exposed, and suggested that secondary reservoirs may be involved in JEV circulation. In the 60s, Gresser et al. discussed the potential implication of domestic birds in the JEV cycle in addition to pigs [175]. More recently, domestic birds have been shown to be exposed to JEV in Nepal and Cambodia, as antibodies were detected by both ELISA and SNT [106,176]. Anti-flavivirus antibodies were also detected in domestic birds in Malaysia, suggesting possible JEV circulation [31]. Experimental studies showed that inoculated ducks and chicken developed different levels of JE viraemia, probably high enough to re-infect mosquitoes [24,177]. 

The question of a non-avian wild reservoir for JEV has never really been assessed, while Southeast Asia hosts the highest wild pig (Suidae family) diversity in the world [178]. As presented above, wild boars were shown to be highly exposed to JEV on two Japanese islands, with the virus genome showing a high homology with JEV that was previously isolated from pigs reared on another Japanese island. The detection of JEV RNA and anti-flavivirus IgG antibodies in wild boars suggested that they could act as additional reservoirs in rural and forest areas [99]. In the north of Queensland, Australia, cross-reacting anti-flavivirus antibodies were detected in almost 80% of feral pigs during a JE outbreak, and the authors suggested that they possible became amplifying hosts [83].

JE was traditionally considered a rural disease. However, several surveys showed that the virus could be transmitted in peri-urban and urban areas, suggesting the presence in these areas of mosquito species able to transmit JEV. That could be the case for *Cx. quinquefasciatus*, which is anthropophilic and competent for JEV transmission [179,180]. *Cx. tritaeniorhynchus*, *Cx. gelidus*, and *Cx. quinquefasciatus* were also trapped in urban households in Vietnam, regardless of whether or not there were pigs in the area [181]. In peri-urban areas, people do not traditionally rear large numbers of pigs or domestic birds. However, increasing urbanization is likely to increase pig numbers on farms, as well as the numbers of farms close to urban areas, thus bringing human and pig JEV-susceptible populations into close proximity with each other. Thus, there is a need to improve our knowledge of the JEV transmission cycle that may not be as simple as we think it is. The existence of secondary reservoirs could explain JEV transmission in areas with no or low pig density, which is the case in some urban or peri-urban areas in Cambodia or Laos for example, or in mountainous areas [108,110,167,182].

JE is primarily a vector-borne disease, but recent surveys suggested that direct transmission from pigs to pigs could occur [21,63]. Two additional in silico studies performed with Cambodian and Hong Kong data showed that incorporating direct transmission in models allowed a better fit to be observed from the serological data than a vector-borne transmission model alone [183,184]. The reemergence of JEV cases in the same locations from one year to another indicates that JEV can overwinter locally [50]. Among others, direct transmission could explain the overwintering of JEV in pig herds in epidemic or endemic regions where JEV transmission and human cases are seasonal [21]. 

Human vaccines are widely used in humans [141,142], and mass vaccination programs were put into place in several countries, such as Japan, South Korea, and Taiwan, where the programs are long-lasting and of high quality, or Nepal, Malaysia, India, Sri Lanka, Thailand, Vietnam, and China, where programs are emerging [1]. As humans are dead-end hosts, human vaccination alone cannot stop virus circulation. Moreover, human cases may re-occur in case of vaccination failure or the emergence of a new strain for which the current vaccine is not effective. JE did re-emerge in South Korea (2010–2015) after a mass vaccination program, which was presumed to have failed to induce lifelong immunity, so older age groups became susceptible again [185]. Vaccines are also expensive, require multiples doses, and remote and/or poor people may not be able to afford them. Additionally, some of these human vaccines may not be 100% effective, as demonstrated in Tandale et al. [186]. 

Pig vaccination could be used as an alternative to control JEV circulation. Vaccinating pigs not only protects them from possible reproductive disorders, but also helps to break the transmission cycle, thus reducing the impact on human health. In northern Bangladesh, Khan et al. built a compartmental model to describe JEV transmission dynamics in this region and to estimate the potential impact of pig vaccination. They showed that vaccinating 50% of the total pig population each year would result in an 82% reduction in the annual incidence of JE infection in pigs [187]. Vaccines are widely used in Japan, Taiwan, and South Korea, where the incidence of disease in swine has been reduced thanks to the use of a live-attenuated strain (Anyang300) conducted throughout these countries for the past 30 years [145,146,188]. 

In epidemic or endemic areas with seasonality patterns, the birth season of pigs has an influence on the age at which pigs get infected by JEV, thereby determining the existence or not of reproductive disorders. In northern Vietnam where the transmission peak occurs between July and September, Ruget et al. used temperature and pig serological data to predict the age of initial JE infection. Pigs born at the end of the winter become susceptible to JEV infection during the period of high circulation and before reaching sexual maturity. On the other hand, pigs born later during the summer are protected by their maternal antibodies during the transmission peak season. They lose their maternal anti-JEV antibodies and thus become susceptible again during winter, when vector abundance is very low. According to authors, around 20% of these pigs did not infected with JEV before 8 months, thus experienced reproduction disorders when getting infected during the following summer. These animals (born between July and September) should be targeted for vaccination [111].

However, pig vaccination has several limitations:(i)The large South Korean vaccination program has not prevented outbreaks in the human population in recent years [145];(ii)Whatever the pig production system and the socio-economical context, the turnover of pig populations is always rapid and therefore the cost of vaccination is high;(iii)JEV may still circulate within vaccinated pig populations [144];(iv)Pig vaccines are based on GIII viruses, the dominant circulating genotype in Asia. However, there are now several studies showing the replacement of GIII by GI that could negatively modify the effectiveness of current vaccines [113,189,190]. JEV was indeed detected in vaccinated populations in aborted fetuses and stillborn piglets in China [68,70]. The authors warned against a potential lack of effectiveness of the vaccine and suggested that the safety of the SA14-14-2 strain belonging to GIII, which is used for vaccine development in pigs, should be reassessed;(v)The currently available vaccines do not confer full protection against the emerging JEV GV strain [191,192].

Finally, the fact that the cycle, at least in some regions, is much more complex (multi-host without or with a weak implication of domestic pig), suggests that a single pig vaccination would not solve the problem directly either [193]. 

Modeling studies were conducted to test new alternative control measures, such as pig herd management. Indeed, the intensity of transmission within a herd depends on the proportion of susceptible and immune animals, and therefore partly depends on the management of the herd. Depending on pig herd size and herd management practices, the proportion of immune pigs within a herd may vary and favor or reduce viral circulation between pigs. For example, the synchronization of piglet birth, which is common in semi-commercial and commercial pig herds, induces regular bursts of susceptible animals when these piglets lose their anti-JEV maternal antibodies. Increasing the duration between two successive litters, which could be controlled through insemination synchronization, would possibly prevent JEV circulation between successive piglet births [194].

As JEV is mainly a vector-borne disease, an alternative to vaccination exists in vector control. Insecticide spray for adults, larvicides such as *Bacillus thuringiensis* toxin for juvenile stages, extracts of *Piper retrofractum* (Piperaceae), or essential oils as oviposition deterrents are techniques that have been used previously [195,196,197]. Another option which is already used in some Cambodian rural areas is to cover pigpens with mosquito nets. The use of insecticide-treated mosquito nets was shown to reduce seroconversion rate in humans and pigs in India [198]. 

In conclusion, this review underlines that JEV epidemiological patterns vary according to the region. Pigs, when present, play a central role, but other hosts may also be involved, such as wild boars or domestic birds. Human vaccination remains the most effective way to protect human populations, but it does not stop the circulation of the virus. It is therefore also necessary to act on the reservoir–mosquito cycle and to adapt existing measures according to the functioning of this multi-host system.

## Figures and Tables

**Figure 1 viruses-11-00949-f001:**
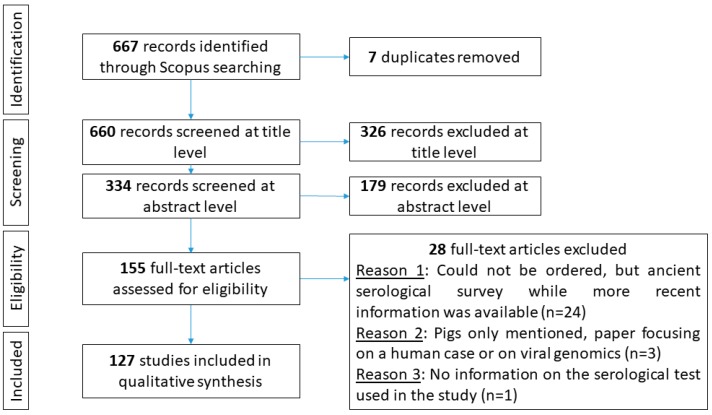
Preferred Reporting Items for Systematic Reviews and Meta-Analyses (PRISMA) flow diagram representing the selection process.

**Figure 2 viruses-11-00949-f002:**
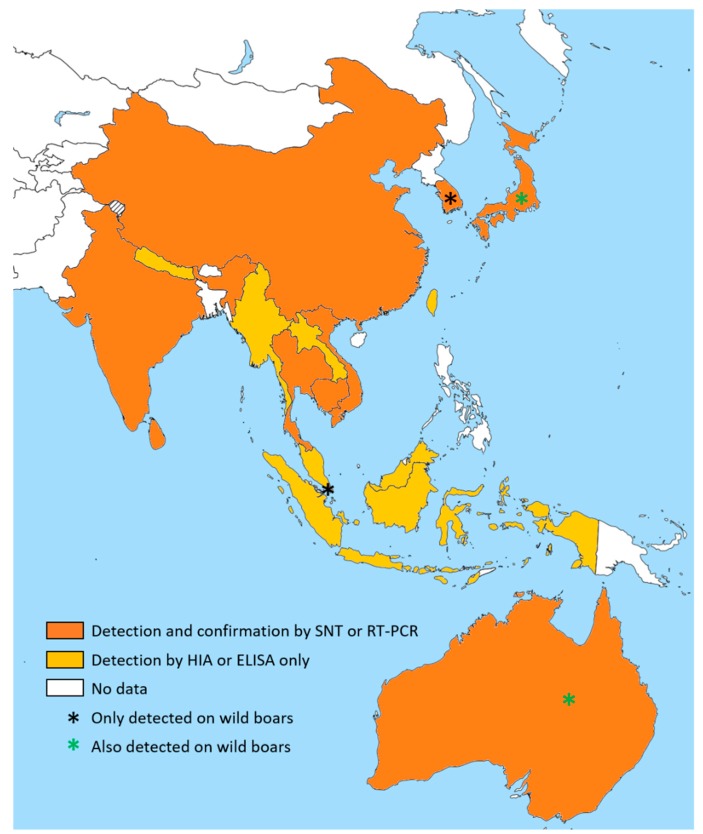
JEV detection in swine from 1966 to 2016, based on selected studies.

**Table 1 viruses-11-00949-t001:** Viraemia and active immune response in pigs after Japanese encephalitis virus (JEV) experimental infection.

Protocol	Viraemia			Immune Response		Reference
	Test	Detection Period	Peak Value	Test	Antibody Detection	
5 sows, IV inoculation, Kanagawa strain, bled daily	ICISM	1–4 dpi	NA	VN in mice	>7 dpi	[51]
4 piglets, SC inoculation, M5/596 and pig9 strains, bled daily	ICISM	For 4 days	2.6 log LD_50_/0.03 mL	HIA	7–35 * dpi	[47]
2 piglets, SC inoculation, 9215 strain, bled daily	ICISM	1–4 dpi	NA	HIA	All positive at 7 dpi	[52]
6 piglets, SC inoculation, Nakayama strain, bled daily	ICISM + RT-PCR	2–5 dpi	NA	SNT	All positive at 14 dpi	[53]
12 piglets, IV + ID inoculation, Nakayama strain, bled daily	RT-PCR	1–5 dpi	10^4 U/mL **	SNT	3–11 * dpi	[22]
10 piglets, IV inoculation, JE-91 strain, bled daily	RT-PCR	3–5 dpi	10^3.5 U/mL **	SNT	All positive at 28 dpi	[54]

IV—intravenous; SC—subcutaneous; ID—intradermal; LD_50_—50% lethal dose; TCID_50_—50% tissue culture infective dose; ICISM—intracerebral inoculation of suckling mice; RT-PCR—reverse transcription-polymerase chain reaction; VN—virus neutralization; HIA—hemagglutination inhibition assay; SNT—seroneutralization test; dpi—day post-infection; *All piglets positive at the date when monitoring ended; ** One U was defined as the viral RNA quantity corresponding to 1 TCID50 of the virus preparation used as the standard by the authors; NA: non-applicable, when viraemia detection was qualitative.

**Table 2 viruses-11-00949-t002:** Sow reproductive disorders after experimental JEV infection and clinical signs and lesions after experimental JEV infection of piglets.

Method	Clinical Signs	Macroscopic Lesions	Microscopic Lesions	Reference
5 pregnant sows, IV inoculation, Fuji strain, bled daily until farrow	No	/	/	[51] 1st experiment
6 pregnant sows, IV inoculation, Kanagawa strain, bled daily until farrow	Mummified and hydrocephalic fetuses in 2/6 litters	/	/	[51] 2nd experiment
14 piglets, IV inoculation, clinical signs monitoring and histopathology after euthanasia	Fever until 4 dpi, depression and hind limbs tremor	No	Non-suppurative encephalitis with perivascular cuffing of mononuclear cells and multifocal gliosis in grey and white matter cerebrum	[62]
10 piglets, intranasal inoculation, clinical signs monitoring and histopathology after euthanasia	Fever until 4 dpi, depression and slight hind limbs tremor on 4 piglets until 10 dpi maximum	No	Non-suppurative encephalitis with perivascular cuffing of lymphocytes, multifocal gliosis, neuronal degeneration, and necrosis	[63]
12 piglets, IV and ID inoculation, clinical signs monitoring, histopathology after euthanasia and RT-PCR on tissues	Fever until 5 dpi, reduce appetite, less manure, and reluctance to move until 6 to 9 dpi	No	Signs of viral meningoencephalomyelitis	[22]
10 piglets, IV inoculation, clinical signs monitoring, histopathology after euthanasia and RT-PCR on tissues	Fever until 5 dpi, mild depression and lethargy until 5 dpi, mild ataxia between 10 and 13 dpi, 2 pigs with hind limb ataxia between 19 and 27 dpi	No	No	[54]

IV—intravenous; ID—intradermal; dpi—day post-infection; RT-PCR—reverse transcription-polymerase chain reaction; /—not investigated.

**Table 3 viruses-11-00949-t003:** Detection of JEV in pig herds with reproductive failure and quantification of reproductive disorders.

Lab Method	Material	Result	Differential Diagnosis Intention	Reproductive Failure	Country	Reference
RT-PCR	Aborted fetuses	Detection	No	No quantitative data	China	[68]
RT-PCR	Sample of 37 CSF of aborted piglets	5/37	No	No quantitative data	China	[69]
ICISM, virus isolation, RT-PCR	Sample of 108 brain tissues of stillborn piglets	20/108	No	No quantitative data	China	[70]
RT-PCR	Sample of 3 brain samples of stillborn piglets	3/3	CSFV, PRRSV, PRV, PPV not detected	30 sows with RF/200 sows (15%)	China	[64]
RT-PCR	31 brain samples of stillborn piglets (all stillborn piglets of the farm)	7/31	No	10 sows with RF/28 sows (36%), 2–5 sb/sow, 31 sb in total	India	[65]
RT-PCR	Sample of 8 brain samples of stillborn piglets	8/8	CSFV, PRRSV, PRV, PPV not detected	37 sows with RF/128 sows (29%)	China	[66]
HIA on body fluids and virus isolation (unspecified method)	Aborted fetuses	Isolation on “some” fetuses	No	50 sows with RF/320 sows (3 farms) (16%)	Japan	[50]

RT-PCR—reverse transcription-polymerase chain reaction; ICISM—intracerebral inoculation of suckling mice; HIA—hemagglutination inhibition assay; CSF—cerebrospinal fluid; CSFV—classical swine fever virus; PRRSV—porcine respiratory syndrome virus; PRV—pseudorabies virus; PPV—porcine parvovirus; sb—stillborn; RF—reproductive failure (abortion or at least one stillborn piglet in the litter).

**Table 4 viruses-11-00949-t004:** Persistence of anti-JEV maternal antibodies in piglets under field conditions.

Protocol	Test	Average Age of Waning of Anti-JEV Maternal Antibodies	Country	Reference
80 piglets, 2–7 months old, from farms, bled monthly	HIA	>4 months old	Japan	[47]
9 piglets, in mosquito traps, bled monthly	HIA	>1.5 months old	Japan	[47]
2 cohorts of 15 piglets, 2 months old, bled every 10 days for 4 months	IgG ELISA	>3 months old	Cambodia, peri-urban	[74]
2 cohorts of 15 piglets, 2 months old, bled every 10 days for 4 months	IgG ELISA	Peri-urban: >2 months oldRural: >3.5 months old	Cambodia, peri-urban and rural	[75]
5 piglets, 2 months old, bled every month for 3 years	HIA	>2 months old	South India	[61]

HIA—hemagglutination inhibition assay; ELISA—Enzyme-Linked Immunosorbent Assay.

**Table 5 viruses-11-00949-t005:** Experimental evidence of JEV direct transmission between piglets.

Infection Route	Test for Viraemia and Oro-Nasal Fluids	Clinical Signs	Viraemia	Oro-Nasal Shedding	Reference
Contact with infected pigs in vector-free buildings	RT-PCR	Yes	Yes, 3–10 dac ~ 10^4^ U/mL	Yes, 5–10 dac ~10^1.5^ U/mL	[21] 1^st^ experience
Oro-nasal inoculation	RT-PCR	Yes	Yes, 1–9 dpi ~10^3.5^ U/mL	Yes, 3–9 dpi ~10^3^ U/mL	[21] 2^nd^ experience
Intranasal inoculation	/	Yes	/	/	[63]

dpi—day post-infection; dac—day after contact; RT-PCR—reverse transcription-polymerase chain reaction; /—not investigated.

**Table 6 viruses-11-00949-t006:** Detection and seroprevalence of JEV in swine in the world.

Sampling Region	Sampling Year	Origin of Sampled Animals (A/F if Pigs)	Age of Sampled Animals	Tested Animals	Serological Test	Anti-JEV Antibodies Evidence	Confirmation Test and Result	Reference	Older References
Australia (T.)	1995	F	NS	90	HIA	63/90	SNT +	[34]	
Australia (T.)	1998	Feral pigs	NS	113	HIA	90/113	nd	[83]	
Cambodia	2007	A and F	~4.3 m (20 d–12 m)	505	HIA and IgG ELISA	65.7% and 63.5%	nd	[80]	[74,75]*
Hong Kong	1968	F	NS	558	HIA	60,4%	nd	[84]	
India	2014	F	>3 m	51	IgG ELISA	35/51	RT-PCR +	[85]	[86,87,88]
Indonesia	2015	F	NS	80	IgG ELISA	32/80	nd	[89]	[90,91]
Japan (M.)	2008	Wild boars	NS	36	SNT	30/36	SNT +	[92]	[93,94]
Japan (Is.)	2010	F	NS	128	HIA	3.1%	RT-PCR -	[95]	[96,97]**
Japan (Ir.)	2010	Wild boars	NS	117	HIA	44.4%	RT-PCR -	[95]	[98,99]
South-Korea	2011	Wild boars	NS	288	SNT	66%	SNT +	[100]	
Laos	2009	A	4–12 m	727	HIA	74.7%	nd	[101]	
Malaysia	2016	F	NS	90	IgG ELISA	40/90	nd	[31]	[102]
Myanmar	1999	F	NS	36	HIA	12/36	nd	[103]	
Nepal	2010	F	4–48 m	454	IgG ELISA	16.7%	nd	[104]	[105,106]
Singapore	1999	Wild boars	NS	28	HIA	28/28	SNT +	[107]	
Sri Lanka	1988	F	1–24 m	951	SNT	32.6%	SNT +	[82]	
Thailand	1983	A	4–12 m	100	HIA	74%	ICISM +	[81]	[13]
China (Tibet)	2015	A	1–6 m	102	IgM ELISA	nr	RT-PCR +	[108]	[109,110]
Vietnam	2010	A	4–8 m	641	IgG ELISA	60.4%	SNT +	[111]	[67,112,113,114,115]
Taiwan	1966	F	3–8 m	6000	HIA	37.3%	nd	[116]	

d—days; m—months; T.—Torres strait, Australia; M.—Main; Is.—Ishigaki; Ir.—Iriomote (Japanese islands); A—abattoir; F—farm; NS—not specified; nr—not relevant as authors looked for IgM; nd—not done. * more recent longitudinal studies—no measured JEV seroprevalence but JEV confirmed by RT-PCR; ** JEV confirmed by either RT-PCR, ICISM, or SNT.

**Table 7 viruses-11-00949-t007:** Detected seasonality in JEV infection in pigs.

Country/Region	Sampling Protocol (Longitudinal or Cross-Sectional (Abattoir or Farm))	Detected Seasonality in Pig Infection	Corresponding Climate	References
Cambodia	Longitudinal^1^	All year	Not relevant **	[74,75]
North Vietnam	Abattoir ^2^	February–October *	Summer/rainy season	[114]
		January–October *		[112]
South Vietnam	Farms ^1^	All year	Not relevant **	[67]
Laos	Abattoir ^1^	June–July *	Summer/rainy season	[101]
		August *		[126]
Malaysia	Abattoir ^2,3^	November–January *	Summer/rainy season	[102]
Indonesia	Longitudinal ^2,4^	October–March *	Summer/rainy season	[125]
Thailand				
Bangkok	Longitudinal ^2,1^	February–May*	Hot and dry season	[14,123]
Chiang Mai	Longitudinal ^2,3^	May–July	Hot/rainy season	[124,129]
Taiwan	Longitudinal ^2,4^	March–October, peak in August/July*	Summer monsoon	[73,78,116]
Japan, main Island	Abattoir ^1,5^	July–November, peak in August/September	Summer (rainfall) to fall (typhoons)	[127]
	Abattoir ^2^	June–December		[50]
	Abattoir ^2,4^	July–August		[118]
	Abattoir ^2^	May-March, peak in August		[120]
Japan, Okinawa	Abattoir ^2^	August	Tropical climate	[121,128]
	Abattoir ^2,4^	April–October		
North East India	Longitudinal ^2^	June–August	Monsoon	[117]
Sri Lanka				
Dry zone	Longitudinal ^3^	October–November	Hot/rainy season	[82]
Wet zone		March–April		
North Australia	Longitudinal ^2,3^	February–April	Hot/rainy season	[83]

* Period of highest detected circulation, but circulation all year around; ^1^ELISA; ^2^HIA; ^3^SNT; ^4^ICISM; ** Not relevant because there is no seasonality.

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
