# Peer review of "How Central Is the Domestic Pig in the Epidemiological Cycle of Japanese Encephalitis Virus? A Review of Scientific Evidence and Implications for Disease Control"

_viruses, 2019, doi:10.3390/v11100949_

Round 1
Reviewer 1 Report
The manuscript by Ladreyt and colleagues presents a useful review summarizing field and experimental aspects of JEV infection in pigs and their role in the epizootiology of this disease. Overall, the authors do a nice job discussing relevant topics for this subject. The manuscript does require considerable revision relative to English language usage and the use of certain virologic terms, and would benefit greatly by having an English-fluent scientific editor assist in revisions. Examples (not inclusive) of items that should be revised and clarified are listed below.
Line 15: remains remains the [leading] cause of human encephalitis
Line 24: Suggest “Vaccination of people does not contribute to suppressing virus circulation, but is clearly the most important means of preventing human disease.”
Line 25: “Implication” seems to be the wrong word here – perhaps importance to transmission?
Line 44: To date, five genotypes (GI to GV) [have been] described and most of isolated strains belong to [GI, GII and GIII].
Line 85: The same person [conducted all of the initial search and screening].
On line 111, you indicate that “It has been experimentally shown that the lowest pig viraemia for Cx. tritaenyorhynchus to reinfect was about log 1.0 and that the reinfection rate was maximum with a viraemia of log 2.5 [46].” I was not able to access reference 46, but would be extremely surprised if mosquitoes could be infected by feeding on a pig with a viremia of 10 [whatever unit of infectivity] per ml; even if the mosquito took up 10 ul of blood, that would contain only 0.1 infectious doses. This is an example of simply regurgitating information from a paper published in 1954 without any evidence of critical interpretation – for example, what kind of assay was used and how sensitive was it compared to modern assays. Another example of non-critical or non-informative statement is found on lines 120-123 (and elsewhere) when viremia measured by infectivity assays is compared to qPCR – in a review such as this, there should be some indication that copies of RNA is significantly higher than infectivity due to defective viruses that are not infectious but have genomes that are detected by PCR. It would be useful to make such a statement, despite the fact that you may not be able to make precise quantitative comparisons.
Table 1. It would be highly useful to have a column that describes peak viremia in addition to the duration of viremia. Additionally, there are several descriptions that are not sensible [e.g. “pig9 strains at 10-32.106 LD50”]; indicating a dose in the form of TCID50/ml (a concentration) is not useful without stating how much of that virus was inoculated.
Line 131: What are “transversal” studies? See also line
Line 136-7: You state “In both studies, piglets were followed for several months, and tested daily by SMIC.” First, SMIC is a unit, not a procedure (should be something like “tested by IC inoculation of suckling mice”. Second, I very much doubt that blood samples were collected daily for several months for testing – please double check this statement.
Line 143: Are you certain that antibodies were detected 3 days after inoculation? This is VERY early to detect an antibody response.
Line 145: you define HIT in the footnotes for Table 1 but should probably spell it out here also.
Table 2: what does “/” signify compared to “No”?
Line 198: could detect (not detected)
Table 4 title: What does “Remanence” mean – that is a physics term related to magnetism, but perhaps has another meaning?
Line 220: 10 to 104 geq-TCID50/ml – do you mean geq/ml OR TCID50/ml?
Line 250+: the phrases “on one hand” and “on the other hand” are jargon and should be replaced.
Line 262: What do you mean or what are you trying to indicate by “when only HIT or ELISA tested”
Line 282: Australia has many flaviviruses – that statement about anti-flavivirus antibodies would seem to have little significance to this review.
Author Response
Comments and Suggestions for Authors
The manuscript by Ladreyt and colleagues presents a useful review summarizing field and experimental aspects of JEV infection in pigs and their role in the epizootiology of this disease. Overall, the authors do a nice job discussing relevant topics for this subject. The manuscript does require considerable revision relative to English language usage and the use of certain virologic terms, and would benefit greatly by having an English-fluent scientific editor assist in revisions. Examples (not inclusive) of items that should be revised and clarified are listed below.
Thank you for your review and comments, please find below the detail of the answers.
Line 15: remains remains the [leading] cause of human encephalitis
Replaced, thank you.
Line 24: Suggest “Vaccination of people does not contribute to suppressing virus circulation, but is clearly the most important means of preventing human disease.”
Agreed, thank you.
Line 25: “Implication” seems to be the wrong word here – perhaps importance to transmission?
Replaced by “involvement”.
Line 44: To date, five genotypes (GI to GV) [have been] described and most of isolated strains belong to [GI, GII and GIII].
Line 85: The same person [conducted all of the initial search and screening].
Replaced, thank you.
On line 111, you indicate that “It has been experimentally shown that the lowest pig viraemia for Cx. tritaenyorhynchus to reinfect was about log 1.0 and that the reinfection rate was maximum with a viraemia of log 2.5 [46].” I was not able to access reference 46, but would be extremely surprised if mosquitoes could be infected by feeding on a pig with a viremia of 10 [whatever unit of infectivity] per ml; even if the mosquito took up 10 ul of blood, that would contain only 0.1 infectious doses.
I replaced with other references where units were clearer: “It has been experimentally shown that viraemia levels of about 10^4 infectious units per ml appeared to be sufficient to transmit the virus to mosquitoes (46–49).”
This is an example of simply regurgitating information from a paper published in 1954 without any evidence of critical interpretation – for example, what kind of assay was used and how sensitive was it compared to modern assays. Another example of non-critical or non-informative statement is found on lines 120-123 (and elsewhere) when viremia measured by infectivity assays is compared to qPCR – in a review such as this, there should be some indication that copies of RNA is significantly higher than infectivity due to defective viruses that are not infectious but have genomes that are detected by PCR. It would be useful to make such a statement, despite the fact that you may not be able to make precise quantitative comparisons.
Thank you for your comment. We did not try to compare viraemia measured by infectivity assay to qPCR, but rather to present the results of the different studies.
As you suggested it, I added the following statement: “Both measures of viraemia cannot be quantitatively compared, as infectivity assays detect the presence of infectious virions, whereas qRT-PCR detects RNA from both non-defective and defective virions. (53).”
Table 1. It would be highly useful to have a column that describes peak viremia in addition to the duration of viremia. Additionally, there are several descriptions that are not sensible [e.g. “pig9 strains at 10-32.106 LD50”]; indicating a dose in the form of TCID50/ml (a concentration) is not useful without stating how much of that virus was inoculated.
I added a column with the values of the detected peak of viraemia, when available as in some studies the information was only qualitative. As the present article did not focus on JEV different strains, I removed the doses of the inoculum when was indicated.
Line 131: What are “transversal” studies? See also line
Transversal studies are cross-sectional serological studies where animals are collected only once, at a single time. We used “transversal” in opposition to “longitudinal” studies where animals are monitored (bled several times, followed). I replaced “transversal” by “cross-sectional” throughout the manuscript.
Line 136-7: You state “In both studies, piglets were followed for several months, and tested daily by SMIC.” First, SMIC is a unit, not a procedure (should be something like “tested by IC inoculation of suckling mice”.
I simplified “SMIC inoculation” that I could read in several papers (as in Williams et al, 2001). Sorry for that, I replaced it by “IC inoculation of suckling mice” (=ICISM) as you suggest.
Second, I very much doubt that blood samples were collected daily for several months for testing – please double check this statement.
In Ueba et al, 1972, piglets were sampled and tested by SMIC inoculation from once a day to once every four days for two months.
Sorry for the mistake concerning the other paper: In Okuno et al, 1973, sentinel piglets were put in two groups on the 2 of July and bled weekly. All sera were tested by HIA and stored at -60°C for 6 months. The last sera collected before observed HI antibody conversion (n=38) were then IC inoculated to suckling mice (i.e. sera collected on the 2nd of July, 9th of July and 16th of July). 5/10 piglets of group 1 were viremic on the 9th of July, and 6/9 piglets of group 2 were viremic on the 16th of july. No information can be given on the duration of detected viremia as sera were not tested again after being shown positives. For the 5 piglets of group 1 though, viremia may have started between the 3rd of July and the 9th) = between 1dpexpo and 7dpexpo. For the 6 piglets of group 2, viremia may have started between the 10 of July and the 16 (i.e. between 8dpexpo and 14dpexpo).
I removed this last reference that did not belong to this section.
Line 143: Are you certain that antibodies were detected 3 days after inoculation? This is VERY early to detect an antibody response.
See figure 6 Ricklin et al 2016: 2 out of 12 experimentally infected piglets “Already at 3 days pi two pigs showed a PRNT50 titer of 10”.
Line 145: you define HIT in the footnotes for Table 1 but should probably spell it out here also.
Corrected, thank you.
Table 2: what does “/” signify compared to “No”?
I replaced “/” by “ns” for “not sought”, as authors did not look for this parameter, unlike when it is indicated “No”.
Line 198: could detect (not detected)
Thank you.
Table 4 title: What does “Remanence” mean – that is a physics term related to magnetism, but perhaps has another meaning?
I replaced it by “Persistence”.
Line 220: 10 to 104 geq-TCID50/ml – do you mean geq/ml OR TCID50/ml?
Park et al, 2018: “For each reaction, a standard curve was generated by 10-fold serial dilution of RNA extract derived from a JEV stock of known titer at 8.52 log10TCID50/ml. Results were reported as geq-TCID50/ml.”
I replaced the unit by “RNA units/ml”.
Line 250+: the phrases “on one hand” and “on the other hand” are jargon and should be replaced.
Thank you, I changed the sentence.
Line 262: What do you mean or what are you trying to indicate by “when only HIT or ELISA tested”
It was not clear sorry, as the parenthesis was only for “anti-flavivirus antibodies”. I wanted to remind that one can only conclude on anti-flavivirus antibodies presence when sera are only tested by HIT or ELISA that are not JEV-specific tests. But this has been explained previously so I will deleted the parenthesis.
Line 282: Australia has many flaviviruses – that statement about anti-flavivirus antibodies would seem to have little significance to this review.
I agree, but we wanted to question the fact that, since anti-virus antibodies have been detected on wild boars around a confirmed outbreak of JEV, some of these antibodies could be anti-JEV. I rephrased so as not to imply that these anti-flavivirus antibodies (on wild boars) are almost certainly anti-JEV antibodies, but that there still may be a link between wildlife and EJ circulation.
Submission Date
30 August 2019
Date of this review
13 Sep 2019 06:44:33
Reviewer 2 Report
General suggestions
This manuscript tried to review how critical is the domestic pig and what the implication is for Japanese encephalitis (JE) control. They provide a systematic review of 127 publications linked to JE virus in swine by using the PRISMA. They suggested pigs are central but the other secondary hosts cannot be ruled out. It is still necessary to identify other potential hosts and to measure the effect of JE control targeting amplifying hosts and other reservoirs in different epidemiological cycles of JE virus. Overall, the comprehensive field and experimental studies were included in this review, and the discussion section was logically presented. However, in the results, too many paragraphs began with four articles, six articles, one study described… etc, and it affected the smooth of the article.
Major comments
While reviewing JEV epidemiology and transmission patterns, some individual characteristics of JEV infection in pigs could be discussed together. The overwintering mechanism could be included. After genotype replacement, the role of pigs and other hosts could be included in the discussion section.
Minor comments
Line 140: “Humoral or Antibodies response of pigs after JEV infection” may be more appropriate. Line 143: Is short viremia associated with the early appearance of antibodies in plasma? Line 144-146: Is it possible that 3-year persistent antibodies in the piglets result from a boosted infection every year? Line 161: what do lines 1 and 2 mean? Please explain “/” in the footnotes of Table 2, 5, and 7 Line 206: Maternal antibodies were more commonly used in JEV studies. Line 224: Ricklin et al found JEV RNA-negative in urine of experimentally infected pigs. Line 227: JEV transmission patterns, epidemiology, and pig-related JEV control Line 228: mechanisms Table 5: Make sure dpi stands for day post infection for the infection route by contact with infected pigs in vector-free buildings. Line 291: Table 6? Confirmation test? Line 315-318 and table 7: Why do you think JE in Taiwan is an endemic pattern with seasonality, where circulation also occurs all year round? Line 337: the main vectors of JEV Line 368: Correct “bigger” litters Line 539: delete “does” it does not
Author Response
Comments and Suggestions for Authors
General suggestions
This manuscript tried to review how critical is the domestic pig and what the implication is for Japanese encephalitis (JE) control. They provide a systematic review of 127 publications linked to JE virus in swine by using the PRISMA. They suggested pigs are central but the other secondary hosts cannot be ruled out. It is still necessary to identify other potential hosts and to measure the effect of JE control targeting amplifying hosts and other reservoirs in different epidemiological cycles of JE virus. Overall, the comprehensive field and experimental studies were included in this review, and the discussion section was logically presented. However, in the results, too many paragraphs began with four articles, six articles, one study described… etc, and it affected the smooth of the article.
Major comments
While reviewing JEV epidemiology and transmission patterns, some individual characteristics of JEV infection in pigs could be discussed together. The overwintering mechanism could be included. After genotype replacement, the role of pigs and other hosts could be included in the discussion section.
Thank you for the review and comments.
Minor comments
Line 140: “Humoral or Antibodies response of pigs after JEV infection” may be more appropriate.
Thank you for the suggestion, I replaced it.
Line 143: Is short viremia associated with the early appearance of antibodies in plasma?
Few articles discussed this assumption. However, Ueba et al, 1972, reported very visually that viraemia did indeed drop as soon as HIT antibodies were detected. This may suggest that viraemia is very short due to the early onset of the immune response.
Line 144-146: Is it possible that 3-year persistent antibodies in the piglets result from a boosted infection every year?
This was not discussed in the article (Geevarghese et al, 1994), but yes, this long persistence might be due either to annual infections boosting immunity. I added a phrase in the text.
Line 161: what do lines 1 and 2 mean?
Line 1 and 2 of table 2 present results of experimental infection of sows on reproductive performance, unlike the rest of the table, which present the impact of experimental infections of piglets.
I removed it from the table title and explained the “/” in the footnotes.
Please explain “/” in the footnotes of Table 2, 5, and 7
Thank you, I replaced and added footnotes.
Tables 2 and 5: “/: not sought”
Table 6: / replaced by nr, footnote: “nr: not relevant as authors looked for IgM”
In table 7: / replaced by nr, footnote: “nr: not relevant because there is no seasonality and therefore no corresponding climate.”
Line 206: Maternal antibodies were more commonly used in JEV studies.
Thank you, I replaced it.
Line 224: Ricklin et al found JEV RNA-negative in urine of experimentally infected pigs.
Sorry for the mistake. I replaced in the text: “Ricklin et al detected JEV RNA in urine at a low frequency (in one urine sample of the 28 monitored piglets) (20), and authors suggested tropism differences between JEV and other flaviviruses as WNV or DENV (74,75).”
Line 227: JEV transmission patterns, epidemiology, and pig-related JEV control
Thank you, changed.
Line 228: mechanisms
Thank you.
Table 5: Make sure dpi stands for day post infection for the infection route by contact with infected pigs in vector-free buildings.
Ricklin et al used “dpi” for needle-infected, contact-infected and oronasally infected piglets (figures 1, 3 and 4 of the Nature communication, 2016).
I replaced dpi by dac (day after contact) in the “Contact with infected pigs in vector-free buildings” line of Table 5.
Line 291: Table 6? Confirmation test?
I removed the question mark.
Line 315-318 and table 7: Why do you think JE in Taiwan is an endemic pattern with seasonality, where circulation also occurs all year round?
Thank you for your question.
Chan et al, 1968, tested two groups of 3000 pigs collected during the two different seasons. Group 1: born between March and August (summer covered), tested in November (3-8 months old). Group 2: born between November 66 and February 67, tested in April (2-5 months old). Group 1=56% HIT+/ Group2=19% HIT+.
Month by month information: Group 1: older pigs = pigs that saw the summer are more + (age effect or season effect?); group2: younger pigs are more positives.
This cross-sectional study led on two different seasons, shows high circulation during summer, but still low circulation in winter.
The two other references (Hurlbut, 1964 and Okuno et al, 1688) are seroconversion studies led during summer, which indeed cannot be used to conclude on a circulation all year around.
I only let the Chan et al, 1968, reference to justify this statement.
Line 337: the main vectors of JEV
Thank you, changed.
Line 368: Correct “bigger” litters
Replaced by “larger”.
Line 539: delete “does” it does not
Thank you, corrected.
Submission Date
30 August 2019
Date of this review
16 Sep 2019 15:57:53
Round 2
Reviewer 2 Report
Line 367-370: I'm not confident about the circulation of JEV all year round in Taiwan. Was seroconversion detected in pigs of group 2 in Chan et al, 1968 study? If not, there is a possibility that HIT+ in younger pigs was maternal antibodies. Was there HIT+ pig aged > 3 months (weaned maternal antibodies) in group 2?
Line 332: JE circulation
Author Response
Thank you for your question.
Chan et al, 1968 underwent a cross-sectional survey. No seroconversion was detected in group 2 as pigs were sampled and tested once.
However, pigs were 2 to 5 months old when tested, as they were born between November 66 and February 67 (summer season not covered) and tested in April 67.
Authors reported that in this group (2), 5 months old pigs had a positive rate of 15.6% (HIT positives), showing that this prevalence cannot be of maternal origin. ("Positive rate is a little higher in swine born in February (24.4%) than those born in November (15.6 %)").
This is why we concluded that there was a circulation all year round.
Line 355, I added :
"In Taiwan, it is believed that the circulation of JE also occurs in winter since Chan et al, 1968, detected HIT antibodies in 15.6% of 5-month-old pigs raised in winter, suggesting that this prevalence is not of maternal origin but due to a slight circulation (116)."